



# Number and size-controlled rainfall regimes in the Netherlands: physical reality or statistical mirage?

Marc Schleiss[1]

[1]Dept. of Geoscience and Remote Sensing, Delft University of Technology

**Correspondence:** Marc Schleiss (m.a.schleiss@tudelft.nl)

**Abstract.**

An experimental study aimed at identifying special rainfall regimes with the help of co-located disdrometers is performed. Eight potentially special events (i.e., 4 number-controlled events and 4 size-controlled events) are identified and examined. However, the cross-check with additional, independent radar measurements reveals no clear evidence of such regimes. The

research underscores the difficulty in confirming seemingly straightforward questions about rainfall patterns and dynamics that have been theorized in the literature for several decades. It also questions the reliability of previous claims and serves as a reminder to approach such problems with more caution, emphasizing the need for rigorous uncertainty analysis and multiple cross-checks between sensors to avoid misinterpretation.

## 1 Introduction

The study of raindrop size distributions (DSDs) is crucial for understanding the microphysical processes involved in rainfall. The DSD is defined as the average concentration N(D) of raindrops of diameter $D$ (mm) per unit volume of air:

$$N(D) = N_T f(D), \tag{1}$$

where $N_T = \int_0^\infty N(D)dD$ (in m$^{-3}$) is the total number of raindrops and $f(D)$ (mm$^{-1}$) is a probability density function for the drop diameters such that $\mathbb{P}[D \leq x] = \int_0^x f(D)dD$ and $\int_0^\infty f(D)dD = 1$.

The rainfall rate R (in mm/h), directly depends on the DSD through the following equation:

$$R = 6\pi 10^{-4} \int\limits_0^\infty D^3 N(D)v(D)dD, \tag{2}$$

where $v(D)$ (m/s) denotes the average fall velocity of a raindrop as a function of drop diameter (Beard, 1976, 1985).

One question that arises when studying rainfall dynamics is whether there are special moments in time during which f(D) or $N_T$ are approximately constant. The first of these special cases is the so-called "number-controlled" regime, in which the size

distribution f(D) is constant and all the temporal variability of the rainfall rate can be explained by changes in $N_T$. The second special case is the so-called "size-controlled" regime, in which $N_T$ is constant and all the variability in the rainfall rate can be explained by changes in f(D).



Conditions of approximately constant size distributions have been theorized for high rain rates by Hudson (1963), Blanchard and Spencer (1970), Srivastava (1971) and Uijlenhoet et al. (2003). According to these studies, number-controlled regimes are likely to occur during the growth phase of warm rain processes, in intense tropical rainfall, strongly organized midlatitude storm systems or persistent orographic rainfall. In the literature, number-controlled rainfall regimes are often associated with equilibrium DSDs (Srivastava, 1971; List et al., 1987), toward which DSDs should converge over time after being released at cloud base. Several numerical and observational studies have looked at the conditions under which such equilibrium states can be reached, how long they take to emerge, and what their shapes are (Zawadzki and De Agostinho Antonio, 1988; Sauvageot and Lacaux, 1995; Brandes et al., 2004; D'Adderio et al., 2018). For example, Zawadzki and De Agostinho Antonio (1988) showed that DSDs in intense, tropical rainfall in Brazil come close to equilibrium shape. Similarly, D'Adderio et al. (2018) detected many equilibrium DSDs in heavy convective events at intensities between 20 and 40 mm/h. Interestingly, while number-controlled regimes and equilibrium DSDs share similar properties, it is worth pointing out that these two concepts are not perfectly equivalent to each other. Equilibrium DSDs are number-controlled by definition. However, just because two consecutive DSDs share the same f(D) does not necessarily imply that they are at equilibrium. Proportionality between DSDs could also be the result of very steady rainfall production mechanisms or homogeneous space-time structures.

Compared to number-controlled regimes, size-controlled regimes are substantially more elusive and controversial. They have been speculated to occur in stratiform-like drizzle (Rogers et al., 1991) and warm orographic clouds during phases where raindrops are neither created nor destroyed but steadily grow by accretion from cloud droplets (Steiner et al., 2004). Carbone and Nelson (1978) also suggested that size-controlled regimes may arise during the dissipating stage of convective cells, while Gunn and Marshall (1955) think they could arise due to size sorting in periods of strong vertical winds or turbulence. However, these are mostly speculations and very little empirical evidence has been provided so far to support these hypotheses.

## 1.1 Background

So far, only a few methods for detecting number and size-controlled regimes based on disdrometer and radar data have been proposed. Steiner et al. (2004) proposed a method for identifying special rainfall regimes through the relationship between the radar reflectivity Z (in $mm^6$ $m^{-3}$) and the rainfall rate R. They emphasized that in a number-controlled regime, the exponent $\beta$ in the equation $Z = \alpha R^\beta$ should be equal to 1, whereas in a size-controlled regime, $\beta$ should be equal to 1.63. The application of this technique can, however, be rather challenging. The substantial measurement uncertainty that impacts Z-R measurements, along with the reliance on strong modeling assumptions (specifically, the fact that Z-R relationships can be approximated by a power-law) imply that large sample sizes are needed to reliably estimate $\alpha$ and $\beta$. Consequently, obtaining a $\beta$ value of 1 or 1.63 does not inherently imply the presence of a special regime, as these particular values could also arise by chance or as a result of measurement uncertainty.

A second method for detecting special regimes pioneered by Uijlenhoet et al. (2003) consists in separating DSDs into different groups of rain rates for which the single-moment DSD normalization framework by Sempere-Torres et al. (1994) can be applied. This leads to a series of exponents describing the scaling properties of the DSDs as a function of rainfall rate. Using this technique, Uijlenhoet et al. (2003) showed that DSDs tend to shift toward a more number-controlled regime at





higher rainfall intensities. However, a primary concern with the approach, aside from the fact that it also requires substantial sample sizes, is that it combines DSDs from various events into a single group. In other words, it provides only a rudimentary overview of the average, climatological scaling properties of DSDs. While this is valuable for modeling purposes, it is very
different from studying the genuine, dynamical changes in DSDs and rainfall rates within a specific event.

Interestingly, the study conducted by Uijlenhoet et al. (2003) also delves into two heavy rain events, during which they conducted a more detailed analysis of the temporal variations in rain rate and DSDs. According to the authors, at least one of these events displayed signs of potentially being governed by a number-controlled regime. Nevertheless, it is important to note that they did not use any formal statistical tests, and that there seemed to be a modest correlation between drop concentration
and mean drop diameter in this event. This suggests that the underlying rainfall regime may not have been purely number-controlled. Given the considerable measurement uncertainty and the absence of an independent data source, such as a second disdrometer, the question of whether the authors indeed captured a number-controlled regime remains unanswered.

## 1.2   Reality or mirage?

To date, the peer-reviewed literature on size and number-controlled rainfall regimes remains notably scarce and no compelling
observational evidence of an authentic, number or size-controlled regime has been presented. The presented evidence consists primarily of anectodal accounts or relies on highly simplified numerical simulations of DSDs along one-dimensional rain columns. Moreover, the few observational studies asserting the existence of special rainfall regimes all suffer from substantial methodological shortcomings, including: 1) limited sample sizes, 2) a lack of comprehensive uncertainty assessment, and 3) an absence of proper validation through an independent secondary data source. In addition to the issues above, there also
appears to be a lot of ambiguity in the definitions themselves. For example, in most studies, it remains highly unclear whether the definitions of number or size-controlled regimes are framed in the context of a mobile reference framework (commonly known as the Lagrangian perspective) or relative to a stationary observed positioned on the ground (referred to as the Eulerian perspective). While this may sound like a subjective choice, and in theory, either viewpoint is permissible, researchers should articulate their choices more explicitly. This is imperative as the temporal variability of DSDs is likely to exhibit substantial
differences when observed from these two distinct frameworks.

Given the problems mentioned above, I believe that there are at least three main issues that need to be addressed: 1) a clear definition of what constitutes number and size-controlled regimes in a given reference framework, 2) a rigorous methodology for identifying special rainfall regimes based on observational evidence and 3) a proper way to validate results taking into account measurement uncertainty and sampling effects. The validation part holds significant importance, as assertions founded
solely on data from a single sensor are susceptible to a range of statistical artifacts and human misjudgements. In this study, we address these concerns by harnessing the benefits of a newly acquired dataset of co-located DSD measurements from two disdrometers at the Cabauw site for atmospheric research. The cross-check between these two sensors substantially decreases sampling uncertainty and false alarm rates, allowing us to draw much stronger conclusions than in previous studies. The primary research questions under investigation is "Do we see empirical evidence of number or size-controlled rainfall regimes



in the Netherlands?" If yes, what type of events do these regimes correspond to? How frequent are they? How long do they last? And what are the conditions under which they occur?

## 2 Data

### 2.1 Disdrometer data

The DSD data used in this study were collected by two co-located Parsivel[2] (Particle Size and Velocity) optical disdrometers (Löffler-Mang and Joss, 2000; Tokay et al., 2014) at the Cabauw Experimental Site for Atmospheric Research. The data span a period of approximately two and a half years, between January $1^{st}$ 2021 and July $31^{st}$ 2023. The sampling resolution of the data is 1 minute. To assess measurement uncertainty and mitigate biases due to wind effects, the laser beams of the two disdrometers are oriented in perpendicular directions (see Figure 1). The DSD data are not published yet but are available upon request to the author.

### 2.2 Micro-rain radar data

In addition to the disdrometer data, we also use time-height profiles of a vertically pointing micro-rain radar (MRR-Pro by Metek GmbH). The MRR is located within a few meters of the two disdrometers (see Figure 1). The sampling resolution is 10 seconds, with a 35 meter range resolution and a maximum range of 4.5 km. The MRR provides valuable data for understanding the dynamics of rainfall and associated microphysics (Peters et al., 2002; Kim and Lee, 2016; Wen et al., 2016; Wang et al., 2017). The main variables used in this study are the attenuation-corrected reflectivity Z (in dBZ), and rain rate R (in mm/h). In additional to that, the raw Doppler spectrum of the MRR can be used to retrieve the DSD at each range gate through the link between the size and fall-velocity of raindrops. More information about this dataset and how to download the data can be found on the Data Catalog of the KNMI data platform.

### 2.3 Weather data

In addition to the disdrometers and the MRR, we also consider basic 10-minute weather data from a meteorological tower located approximately 150 meters away from the disdrometers. The variables of interest include air temperature, dew point temperature, wind speed and wind direction at 7 different heights from the ground up to 200 meters. However, for this study, only the measurements at the ground level were taken. The weather data are used to provide additional context and better understand the environmental conditions under which the special rainfall regimes detected by the disdrometers occurred.





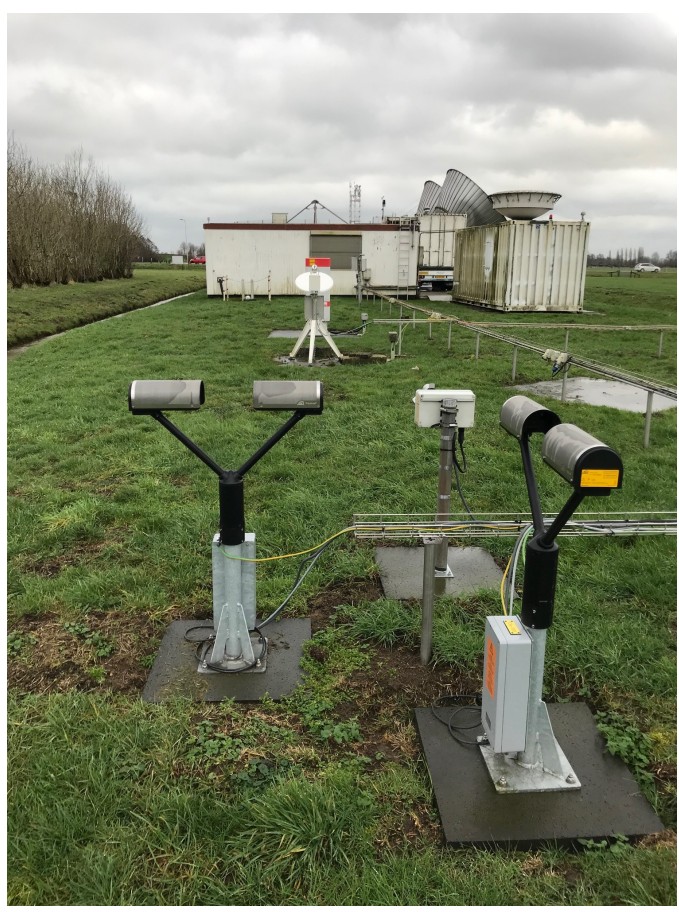

**Figure 1.** Parsivel disdrometers at the Cabauw site for atmospheric research in the Netherlands. The micro-rain radar is visible in the background.

## 3 Methods

### 3.1 Definition

In this paper, all special regimes are defined from the vantage point of a fixed observer on the ground. A number-controlled regime corresponds to a timeframe during which the rainfall rate R, as defined in Equation 2, varies by at least 1 mm/h, while f(D) in Equation 1 remains constant. Conversely, a size-controlled regime is defined as a time period over which R fluctuates by at least 1 mm/h while the total drop number concentration $N_T$ remains constant. The 1 mm/h threshold on the range R is used to avoid trivial cases during which the entire rainfall process is stationary and both $N_T$ and f(D) are constant at the same time. The fact that everything is defined from the perspective of a fixed observer makes it more challenging to link the regimes on the ground to microphysical processes occurring along the horizontal and vertical dimensions, such as drop coalescence and breakup. However, this fixed observation framework is imperative for conducting experimental studies based



on disdrometer data. In the results section, we delve deeper into the implications of this choice and explore in greater detail
how it may influence the conclusions.

## 3.2    Moving time window approach

The proposed method for identifying number and size-controlled regimes is very simple. A 15-minute moving time window is
applied to the 1-min disdrometer data in order to estimate the local (co)-fluctuations in DSD-related quantities. The goal is to
find short time intervals over which $N_T$ or $f(D)$ remain approximately constant. The length of the time window is somewhat
subjective, but experience shows that 15 minutes provides a sufficiently large sample size to capture meaningful temporal varia-
tions in the data, while still maintaining a reasonably high temporal resolution. For completeness, some additional experiments
using shorter time windows of 10 and 5 minutes were also conducted (see Section 4.5). To avoid issues due to intermittency,
we exclude all time intervals in which one or more of the 15 rain rate values are zero. The elimination of dry periods within
the designated time window guarantees equal sample sizes. It also avoids special cases in which two or more consecutive rain
pulses separated by a few minutes of dry weather are erroneously attributed to the same rainfall regime.

## 3.3    Proxy for f(D)

Unfortunately, since f(D) is a distribution, it is not trivial to quantify its temporal changes over the moving window. One
approach would be to calculate the average dissimilarity of f(D) with respect to the average distribution over the time window
using some form of statistical metric such as the Kolmogorov-Smirnov test or Kullback-Leibler divergence. However, such
an approach can be subject to large uncertainties due to the fact that most raindrops are small and that the Parsivel is known
to have issues when it comes to accurately measuring the concentration of small drops. Therefore, a much simpler approach
in which the mass-weighted mean drop diameter $D_m$ and its changes over time are used as a proxy for studying the changes
in f(D) is used. The mass-weighted mean drop diameter $D_m$ puts more weight on the larger drop sizes, which are the most
important for understanding variations in rainfall rates.

$$D_m = \frac{\int\limits_0^\infty D^4 N(D) dD}{\int\limits_0^\infty D^3 N(D) dD} \tag{3}$$

Because it is a ratio between two successive moments, $D_m$ is independent of $N_T$. This means that if f(D) is constant, $D_m$ must
be constant as well. A constant $D_m$ does, however, not necessarily imply a constant f(D), although it is very difficult to imagine
a plausible scenario in which f(D) could change significantly without causing any fluctuations in $D_m$. In case of doubt, one
can always use a second, characteristic drop size in addition to $D_m$, such as the mean or median drop diameter. If multiple
characteristic drop sizes are constant, then f(D) is very likely to be constant.



## 3.4 Moving window statistics

For each 15-minute time window, the following statistics are computed:

1. The coefficients of variation $\text{CV}(N_T)$ and $\text{CV}(D_m)$ of $N_T$ and $D_m$ over the time window, defined as the ratio between the standard deviation and the arithmetic mean.

2. The correlation coefficients $\rho(N_T, R)$ and $\rho(D_m, R)$ over the time window. For $N_T$, the Pearson correlation is used while for $D_m$, the Spearman rank correlation is used. In addition to computing the value of the correlation coefficient, the cor.test() function in R-CRAN (R Core Team, 2023) is used to determine whether the sample correlations are significantly different from zero at the 5% level. Note that the CV and correlations are only computed on time intervals for which all rain rates are strictly positive and R varies by at least 1 mm/h.

## 3.5 Detection rules

Number and size-controlled regimes are detected using the following criteria:

– Time-intervals for which $\rho(N_T, R) \geq 0.9$, $\text{CV}(D_m) \leq c_{D_m}$ and $\rho(D_m, R) = 0$ at the 5% level are labeled as number-controlled.

– Time intervals for which $\rho(D_m, R) \geq 0.9$, $\text{CV}(N_T) \leq c_{N_T}$ and $\rho(N_T, R) = 0$ at the 5% level are size-controlled.

The thresholds $c_{D_m}$ (-) and $c_{N_T}$ (-) can be set by the user to achieve a specific detection sensitivity or dynamically adapted to the characteristics of the data over the time window. In our case, the thresholds were set to be equal to the $5^{\text{th}}$ quantile of the empirical distribution of $\text{CV}(D_m)$ and $\text{CV}(N_T)$ over the entire 2.5 years of data, which roughly corresponds to $c_{D_m}$=0.06 and $c_{N_T} = 0.1$. Experience shows that these thresholds work well for identifying the most interesting periods of almost constant $N_T$ or $D_m$ while allowing for some residual variability due to measurement uncertainty. Note that there is little value in trying to optimize these thresholds since we are not interested in identifying all events but only the most promising cases with the strongest correlations and lowest coefficients of variation.

## 3.6 Cross-check with the other disdrometer

The detection algorithm described above is applied independently to each disdrometer, by shifting the time window by 1 minute at each step. Whenever a number or size-controlled regime is detected, a cross-check with the other disdrometer is performed to assess the reliability of the detection. If both disdrometers detect the same regime within $\pm$ 1 minute, the detection is labeled as "confirmed". The tolerance of $\pm$ 1 minute is used to account for the fact that the two internal clocks of the disdrometers are not always completely syncronized, which can lead to small time differences in terms of the timing of the individual detections.





### 3.7 Grouping into events

In the final step, all confirmed detections separated by less than 45 minutes of dry weather are aggregated into "events". This grouping is only done for visualization purposes and for understanding the environmental conditions under which the regimes occur (e.g., the time of the year, temperature and wind speed). For each event, the data from the vertically pointing micro-rain radar are used to get additional insight into rainfall types, melting layer height (if it exists), vertical variability and DSD dynamics.

## 4   Results

The following describes the results obtained after applying the moving window-based detection method to the full dataset of co-located DSD measurements in Cabauw. In total, 376 time intervals of length 15 minutes with potential number-controlled rainfall regimes and 139 time intervals with potential size-controlled regimes were identified. However, only 11 time intervals with number-controlled regimes and 12 intervals with size-controlled regimes were confirmed during the cross-check with the

other disdrometer. Therefore, the first question we must address is: why are more than 90% of all detected special regimes not corroborated by the other disdrometer? Quality control shows that in general, the two disdrometers agree very well with each other. There are no large biases, time shifts or data gaps that could explain such a high level of disagreement between the two sensors. Therefore, the large false alarm rate must be the result of large measurement noise and sampling uncertainty. Indeed, it is important to keep in mind that both the correlation coefficient and coefficient of variation over each time window are

computed based on only 15 samples. This small sample size, combined with the large measurement uncertainty of the Parsivel disdrometer at 1-minute resolution leads to a lot of statistical issues. For example, the time series of $D_m$ or $N_T$ are likely to exhibit spurious correlations with R due to correlated measurement noise (e.g., wind effects or internal processing/filtering), which can make it look like there is a special rainfall regime. Another serious issue in small sample sizes is the presence of high leverage points, that is, observations that have a large impact on the estimated correlation coefficient or coefficient of variation.

For example, a single, large rainfall value within the 15 minute window can be sufficient to create a high correlation between R and $N_T$ (or $D_m$). Our results show that the cross-check with the second disdrometer is essential in mitigating these statistical issues.

### 4.1   Events with confirmed detections

Once the cross-checks were done, the 23 confirmed detections were grouped into distinct meteorological "events" (see Meth-

ods). This resulted in 4 distinct events with number-controlled regimes and 4 events with size-controlled regimes. The characteristics of these events, such as the average rain rate, drop size, concentration, temperature and wind speed are summarized in Table 1. Figure 2 shows the time series of R, $D_m$ and $N_T$ for the first event of each regime. The other time series for events 2, 3 and 4 can be found in Figures 3 and 4





**Table 1.** List of all number-controlled (NC) and size-controlled (SC) events with confirmed detections.

| | Time | $\bar{R}$ | maxR | $\bar{D}_m$ | $\bar{N}_T$ | $\bar{T}_a$ | $\bar{T}_d$ | $\bar{W}_s$ |
|---|---|---|---|---|---|---|---|---|
| NC1 | 2023-01-11 00:54 – 01:12 | 1.16 | 2.26 | 0.64 | 2163 | 9.8 | 9.8 | 16.7 |
| NC2 | 2023-03-13 00:05 – 00:20 | 1.22 | 1.93 | 0.61 | 1845 | 8.3 | 7.8 | 15.1 |
| NC3 | 2023-04-06 11:53 – 12:08 | 1.23 | 2.07 | 1.05 | 342 | 6.0 | 4.7 | 10.3 |
| NC4 | 2023-07-27 00:04 – 00:24 | 0.83 | 1.85 | 0.87 | 358 | 14.5 | 11.8 | 9.4 |
| SC1 | 2022-06-05 16:08 – 16:25 | 4.13 | 7.41 | 1.64 | 430 | 15.7 | 15.0 | 9.9 |
| SC2 | 2022-12-19 07:57 – 08:14 | 0.87 | 2.02 | 1.06 | 218 | 6.6 | 6.7 | 16.2 |
| SC3 | 2023-01-14 06:52 – 07:11 | 1.28 | 2.42 | 0.94 | 419 | 5.0 | 4.6 | 15.0 |
| SC4 | 2023-03-09 23:18 – 23:34 | 2.13 | 3.56 | 1.32 | 340 | 2.6 | 2.4 | 7.9 |

**Figure 2.** Time series of R, $D_m$ and $N_T$ for the first potential number-controlled regime (NC) and first potential size-controlled regime (SC).



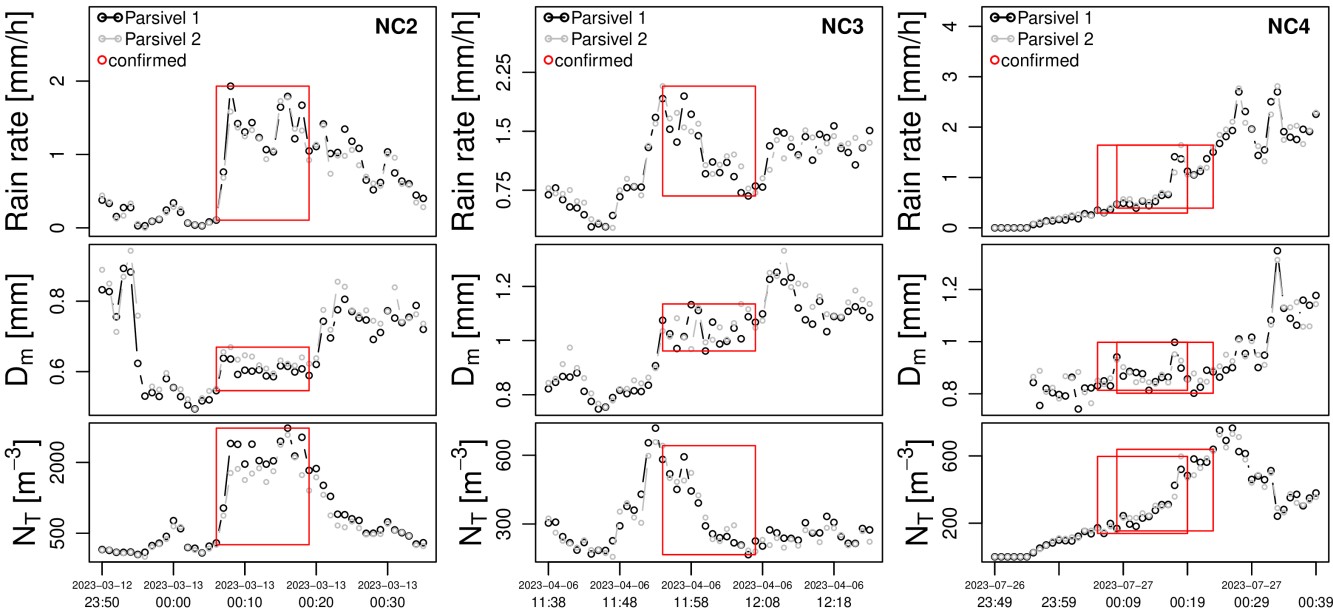

**Figure 3.** Time series of R, $D_m$ and $N_T$ for the second, third and fourth potential number-controlled regime (NC2-NC4).

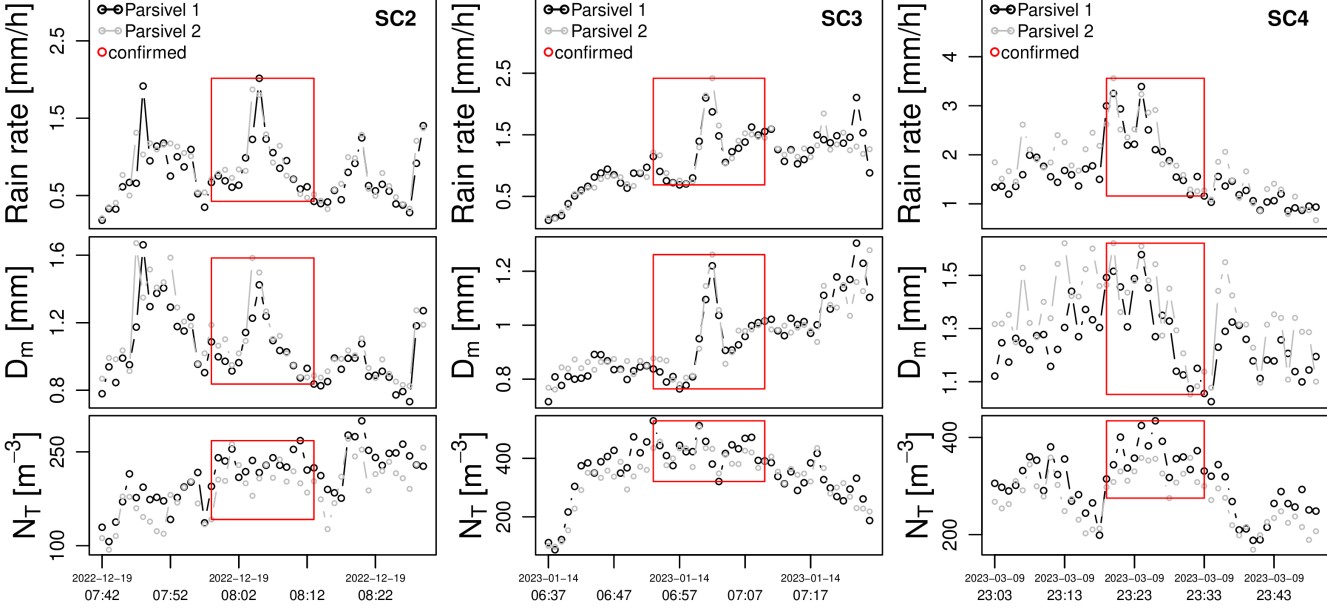

**Figure 4.** Time series of R, $D_m$ and $N_T$ for the second, third and fourth potential size-controlled regime (SC2-SC4).

Looking at Table 1, we can see that all four number-controlled regimes occurred within periods of low rainfall intensities
in the order of 1-2 mm/h. This is very surprising given that in the literature, number-controlled regimes are predominantly





associated with heavy, convective rain. We will come back to this issue shortly. Meanwhile, we can point out that three out of four number-controlled regimes were nocturnal events with small drop sizes and narrow size distributions ($\bar{D}_m$ between 0.61-0.87 mm). NC1-NC2 had relatively large drop number concentrations in the order of 2000 m$^{-3}$ while NC3-NC4 had lower drop counts in the order of 350 m$^{-3}$. No clear seasonal effects could be observed. The events are spread out over winter,

spring and summer and cover a relatively broad range of temperatures (6.0-14.5°C) and wind speeds (9.4-16.7 m/s). In three out of four cases, the number-controlled regime coincided with the most intense part of the event, while for NC4, the regime occured more toward the beginning of the event.

The four events with size-controlled regimes correspond to slightly higher rainfall intensities up to 7.41 mm/h. Nonetheless, the average drop number concentrations remain quite low ($N_T$ between 218 and 430 drops per cubic meter), and the mean

drop sizes are low to moderate (0.94-1.64 mm). This aligns well with the theoretical expectations for size-controlled regimes in the literature and the notion that size-controlled regimes may occur during stratiform rain and drizzle. However, there was no discernable pattern in terms of the timing of the special regimes within the event. For SC1, the size-controlled regime was detected between two consecutive rainfall peaks. For SC2 and SC3, the regime coincided with the rainfall peak, while for SC4, it occurred shortly after the peak. Similarly, no specific seasonal pattern or correlation with temperature or wind speed could

be observed.

### 4.2  Time-height profiles of MRR

Next, the vertical profiles of the micro-rain radar are used to get more insight into the rainfall types and space-time structures associated with the detected regimes. Figures 5 and 6 show the time-height profiles of reflectivity corresponding to the 4 number and 4 size-controlled regimes.

One can see that with the exception of NC1-NC2, all identified events have a clear melting layer signature consistent with stratiform precipitation. As for NC1-NC2, they probably correspond to warm rain events in shallow cumulus clouds which are known to produce a large number of small drizzle droplets through collision-coalescence processes, as highlighted by the small values of $D_m$ and high values of $N_T$ in Table 1. Interestingly, neither of the 8 time-height profiles of reflectivity show any distinct patterns that would suggest the presence of a special rainfall regime. However, since it can be rather difficult to

conclude anything meaningful about DSD dynamics just by looking at time-height profiles of reflectivity, a more in-depth analysis of the MRR data was performed. To gain further insight, the Doppler velocity spectra from the MRR were used to retrieve the DSDs at specific heights, following the procedure described in (Peters et al., 2005; Reinoso-Rondinel and Schleiss, 2021). The retrieved DSDs were then used to calculate key quantities such as $D_m$ and $N_T$ at specific heights.

Unfortunately, the retrieved $N_T$ values from the MRR were too noisy and uncertain to be useful. This a common problem

in radar-based DSD retrieval algorithms, and can be explained by the fact that radar mostly provides information about the higher-order moments of the DSD such as reflectivity (moment of order 6), and very little information about the lower order moments such as $N_T$ (i.e., moment of order zero). Fortunately, the retrieved $D_m$ values, which are independent of $N_T$, were still reasonably good. While $D_m$ alone is not sufficient to get a full picture of DSD variability, it can still be very useful to confirm the presence or absence of special rainfall dynamics. Indeed, for a number-controlled regime, the correlation between




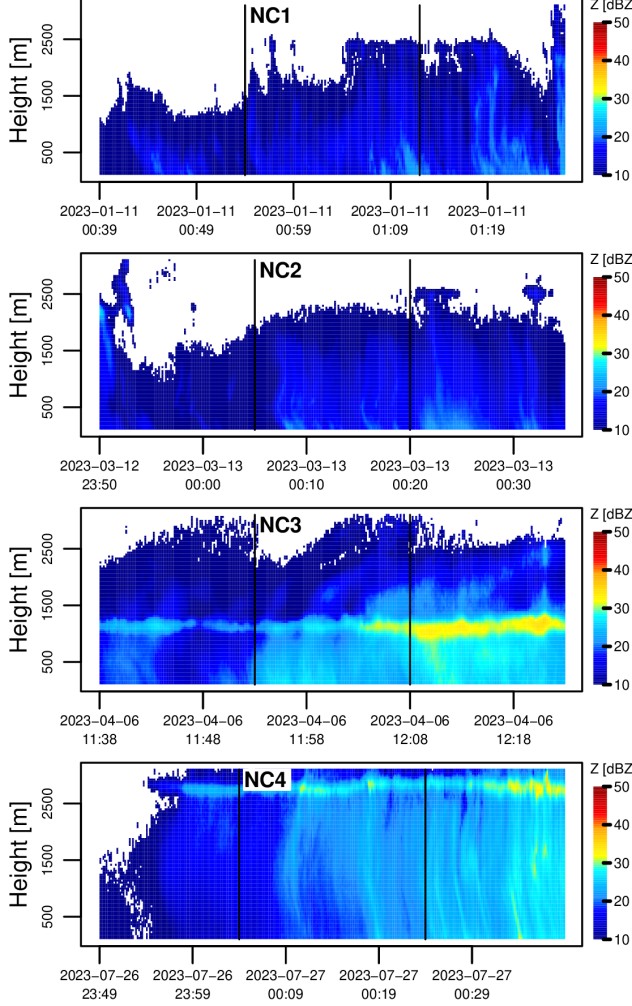

**Figure 5.** Time-height profiles of (attenuation-corrected) reflectivity Z (in dBZ) from the micro-rain radar for the 4 number controlled events identified by the Parsivel disdrometers. The black vertical lines represent the time intervals over which the confirmed detections occurred.

$D_m$ and R should be zero, while it should be 1 for a size-controlled regime. Therefore, the $D_m$ time series from the micro-rain radar can be used as an independent source of information to cross-check the special regimes detected by the disdrometers.

    Figure 7 shows the time series of retrieved $D_m$ values from the MRR at a height of 175 meters above ground for all 8 events. The 175 m level corresponds to the fifth range gate in the MRR, which is the first range gate that is consistently free from ground clutter. The red boxes represent the time intervals over which the Parsivel detected a special regime. The micro-rain

radar has a higher sensitivity to small drops than the Parsivel, which means that the retrieved $D_m$ values can be lower than the ones measured by the Parsivel, especially during drizzle events such as NC1. The retrievals from the micro-rain radar also have higher temporal resolutions (i.e., 10 seconds compared to 60 seconds for the Parsivel). Nonetheless, we can see



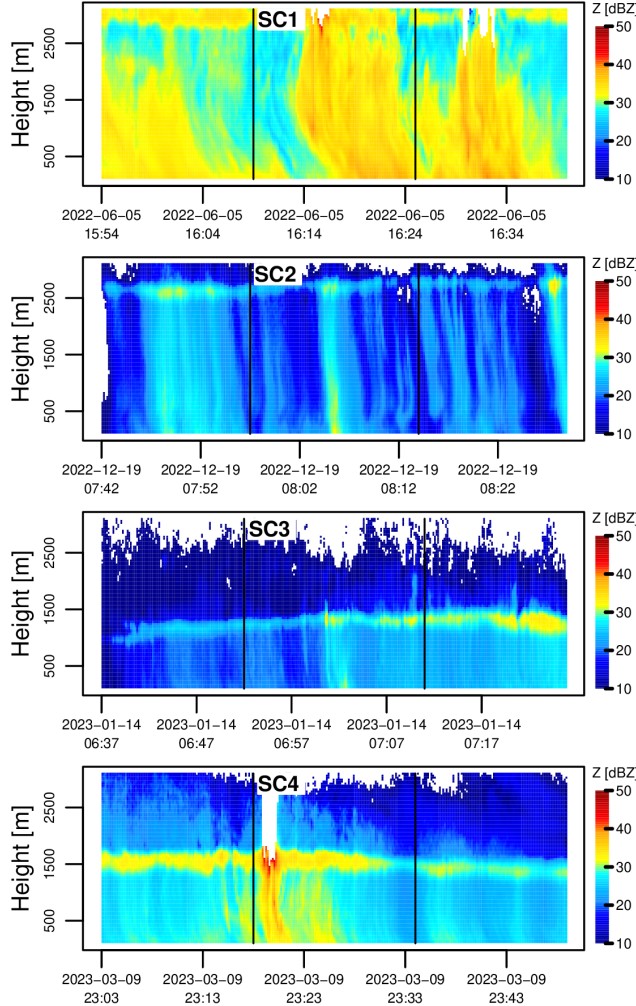

**Figure 6.** Same as 5 but for the 4 size controlled events identified by the Parsivel disdrometers.

that the $D_m$ values retrieved from the micro-rain radar tend to be rather consistent with the ones measured by the Parsivel disdrometers. While this good agreement between the two sensors is encouraging, the correlation values between $D_m$ and R in Figure 7 clearly do not agree with the theoretical, expected values. For the number-controlled regimes, the correlation coefficient between $D_m$ and R should be zero, while for the size-controlled events, it should be 1. This is clearly not the case, except maybe for NC1 and NC4, where one could argue that the small correlation coefficient $\rho(D_m, R) = 0.23$ may be the result of random errors and uncertainties during the retrieval of $D_m$.



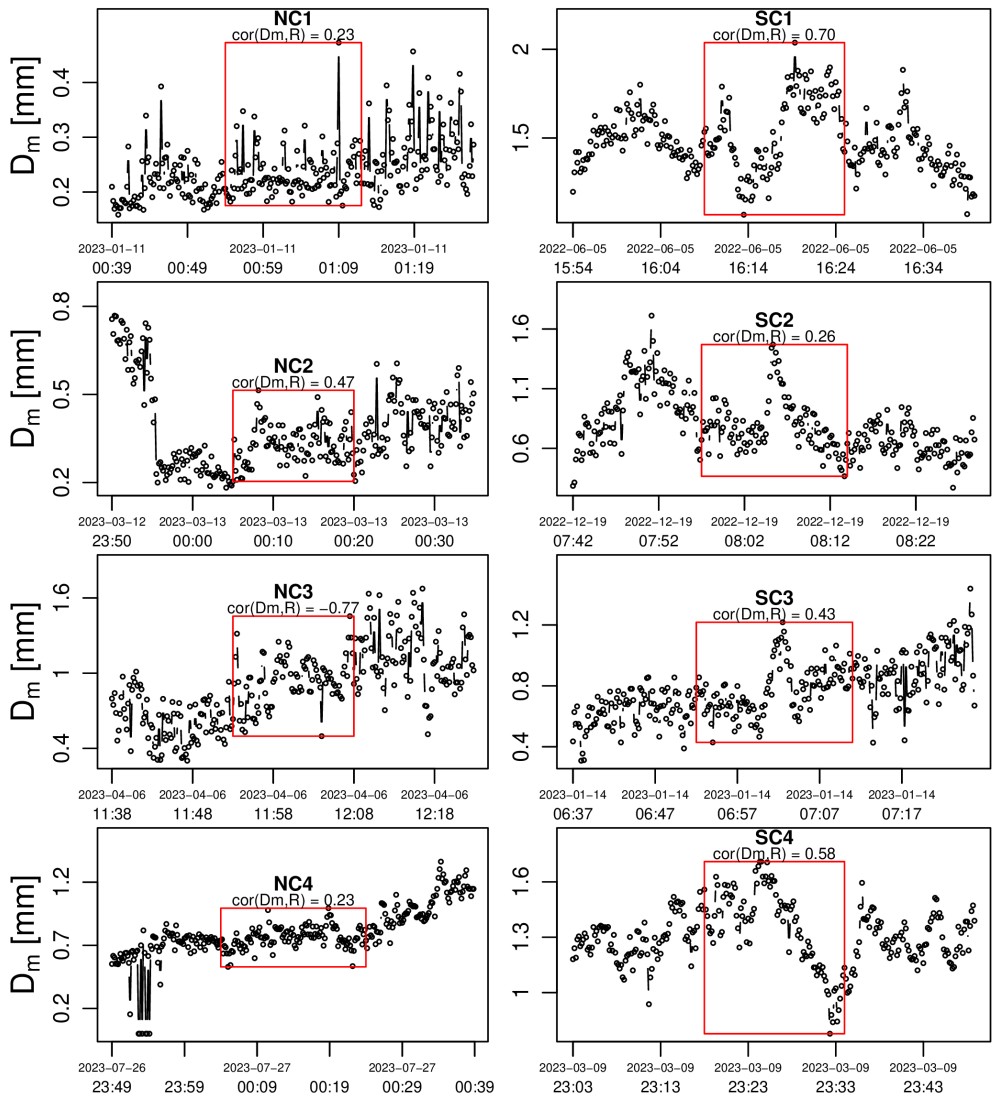

**Figure 7.** Time series of mass weighted mean drop diameter $D_m$ (in mm) retrieved from the micro-rain radar at a height of 175 m above ground, for the 4 number controlled (NC) and size-controlled (SC) events identified by the Parsivel disdrometers. The red boxes represent the time intervals over which the Parsivel disdrometers detected a special regime.

### 4.3 Z-R relations from MRR

Next, the diagnostic method proposed by Steiner et al. (2004) based on the relationship between the radar reflectivity Z and rain rate R is considered. Following the method, we calculated the exponents $\beta$ in $Z = \alpha R^{\beta}$ for each event and compared them to the expected, theoretical values. According to theory, $\beta$ should be equal to 1.0 for a number-controlled regime and 1.63 for a size-controlled regime.





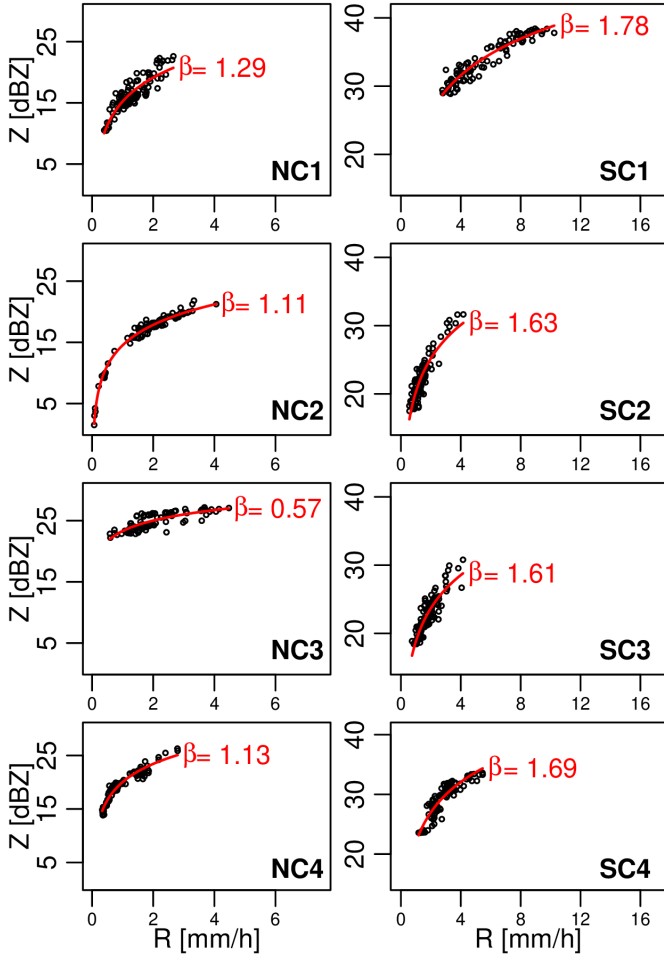

**Figure 8.** Z-R relations at 175 m height for the 8 potential special rainfall regimes. Note that only the Z and R values during the time interval of confirmed detections were taken.

Figure 8 shows the Z-R relations derived from the MRR measurements at a height of 175 meters above ground. Note that the
rain rate values used for this analysis were calculated based on the retrieved DSDs (see above). We can see that in three out of the four size-controlled events (i.e., SC2-SC4), the estimated $\beta$ values agree rather well with the expected, theoretical value of 1.63. However, this result must be interpreted very carefully, since the estimated $\beta$ values have large uncertainties in the order of $\pm$ 0.15 (see confidence intervals in Table 2 for more details). For the number-controlled events, one the other hand, none of the confidence intervals cover the theoretical value of 1.0. This, together with the previous analysis of $D_m$, casts substantial
doubt on the physical reality of the special regimes identified by the disdrometers. It also underscores how different conclusions can be reached based on the choice of rainfall sensor or technique employed to identify the special regimes. Importantly, this conclusion is not limited to the 175 m height, and strong discrepancies between the detections from the MRR and the Parsivel disdrometer can be seen for the other range gates as well.



**Table 2.** 95% confidence intervals for the estimated $\beta$ values in Z-R relationship of the MRR data, at 175 meters above ground, during the 8 special regimes detected by the disdrometers.

|      | 2.5%  | 97.5% |
|------|-------|-------|
| NC1  | 1.159 | 1.431 |
| NC2  | 1.077 | 1.141 |
| NC3  | 0.479 | 0.657 |
| NC4  | 1.072 | 1.183 |
| SC1  | 1.664 | 1.903 |
| SC2  | 1.471 | 1.791 |
| SC3  | 1.459 | 1.770 |
| SC4  | 1.526 | 1.851 |

## 4.4 Interpretation of results

The analyses presented above clearly show the limitations of detecting special regimes on the basis of in-situ DSD measurements only. On one hand, we can confidently say that according to our definitions, and from the perspective of a fixed observer on the ground, the 8 events identified by the disdrometers technically qualify as number and size-controlled regimes. However, the time-height profiles of the micro-rain radar clearly show that none of the 8 identified events contain any special rainfall dynamics worth mentioning. For example, all four events with number-controlled events (NC1–NC4) identified by the dis-

drometers correspond to low intensity rainfall (i.e., stratiform or drizzle), with extremely narrow size distributions centered around small raindrops. For these events, the drop sizes are close to the lower limit of what can be reliably measured with the Parsivel disdrometer. Consequently, the temporal variations in $D_m$ are indistinguishable from measurement noise and $D_m$ therefore appears to be uncorrelated with R. While such a situation technically qualifies as a number-controlled regime according to our definition, it is significantly different from the number-controlled regimes (and equilibrium DSDs) that have been

theorized in the literature.

The four identified events with size-controlled regimes (SC1-SC4) revealed another important problem related to the choice of the reference framework used to define the special rainfall regimes. Because of the horizontal motion of the rainfall field, the temporal variability of the DSDs at the ground may not be representative of the variability within a given rain cell. For example, events SC1, SC2 and SC4 clearly show that whenever multiple rain cells with slightly different rainfall intensities move over

the measurement area, the temporal fluctuations in $D_m$ as seen by an observer on the ground might be enough to trigger the detection of a size-controlled regime, even if none of the individual cells actually contain any special rainfall dynamics.





## 4.5 Sensitivity to the choice of the time window

The final question we need to address is: do special rainfall regimes exist in the Netherlands? Just because we could not convincingly demonstrate their existence in this study does not mean that they do not exist. Another reason why we might not
see any clear evidence of special regimes could be that the 15-minute time window is simply too long. Indeed, D'Adderio et al. (2018) pointed out that most equilibrium DSDs only last for very short time intervals (i.e., 2 to 4 min). This also explains why the only special events detected by our method were long-lasting stratiform or drizzle events with low rainfall intensities and homogeneous spatial structures. To further investigate this, some additional experiments were conducted using smaller time windows of 10 minutes and 5 minutes. Unfortunately, the use of a shorter time window did not turn out to be a good strategy.
The shorter windows lead to more "events" being detected, which makes sense given that it is more likely for $N_T$ or f(D) to remain constant over 5 minutes rather than 15 minutes. However, the cross-check with the MRR showed that none of the new detected events contained any credible evidence of a special rainfall regime. It is also worth pointing out that none of the newly detected number-controlled regimes exceeded 8 mm/h, and that the vast majority of the events were stratiform events and drizzle. This result holds regardless of the chosen time window and suggests that there is no clear benefit in using a smaller
time window. On the contrary, the use of a shorter time window appears to be detrimental to the method, as it leads to higher sampling uncertainty and false alarm rates overall. In the end, it might just not be possible to successfully and reliably detect special rainfall regimes using 1-min DSD data from Parsivel disdrometers, regardless of the time window used.

## 4.6 Should the definition be changed?

Another issue that deserves attention but has never been mentioned so far is the fact that special rainfall regimes are defined
with respect to the rain rate, which is a flux over a fixed area. The problem with this definition, as can be seen in Equation (2) is that the rain rate actually depends on three quantities: $N_T$, f(D) and v(D). Therefore, even if one of the two DSD components (e.g., f(D) or $N_T$) were to be constant, the other component would not be enough to fully explain all the natural variability in R. The fall velocity v(D) of the raindrops, which depends on wind, pressure and temperature, would also have to be considered. This may sound like a minor issue given that in most rain events, the fall velocity of raindrops is fairly well constrained by
their diameter. However, it is a serious issue in heavy convective rain with strong updrafts and downdrafts. In addition to that, it should also be said that accurately measuring fall velocities using optical disdrometers is a challenge in itself, which results in additional uncertainty during the calculation of rain rates. Perhaps, a different definition of number and size-controlled regimes that does not depend on the fall velocity should be considered. For example, a definition that puts more emphasis on liquid water content rather than the rain rate. Liquid water content is proportional to the 3rd order moment of the DSD and
closely related to the rainfall rate. However, it does not depend on the fall velocity and its fluctuations. The only factor that can influence liquid water content, apart from variations in DSD, is the mass density of liquid water, which is very close to constant (in rain). Another advantage of using liquid water content instead of rain rate is that it is less sensitive to microphysical processes such as drop breakup and collisional growth, which preserve the mass in a given volume but not the rain rate. In fact, one could even go as far as to say that many other reference variables besides rain rate and liquid water content could



be used to define additional, special types of rainfall regimes, including higher moments such as the reflectivity Z. Such an approach bears a lot of similarities to the single-moment DSD normalization approach by Sempere-Torres et al. (1994) and double-moment normalization by Lee et al. (2004) in which key moments of the DSD are taken as a reference to study the variations of natural DSDs given some fixed physical quantities. However, while DSD normalization techniques are usually applied on a climatological time scale, by combining the properties of many different DSDs across multiple events, the scaling

properties that are considered in this study only pertain to a short period of time (e.g., a few minutes), and single rainfall cell.

## 5   Conclusions

A systematic search for size and number-controlled rainfall regimes in the Netherlands was conducted based on two and a half years of DSD data collected by two co-located Parsivel disdrometers in Cabauw. A total of 8 promising events were identified. However, upon closer inspection, it became clear that none of these 8 events contained clear, conclusive evidence

of any special rainfall regime. Clearly, what constitutes a special regime lies in the eye of the beholder, and multiple answers are possible, depending on the definition and reference framework used to describe the regimes. While the 8 events identified by the disdrometers technically qualify as number/size controlled from the perspective of a fixed observer on the ground, the MRR data clearly showed that they do not, in fact, correspond to any special microphysical regime.

Subsequent analyses using different thresholds and time windows did not change that conclusion. Based on this, we can

confidently conclude that if special rainfall regimes exist, they likely occur on timescales that are beyond the detection capabilities of the Parsivel disdrometer. Another possibility could be that these special rainfall regimes only manifest at very high intensities (e.g., above 100 mm/h), under specific atmospheric conditions that are not prevalent in the Netherlands, or possibly require a more extended period of observation.

In addition to the issue of insufficient temporal resolution, serious methodological challenges were highlighted. For example,

we showed that because of horizontal motion, a fixed-observer on the ground is likely to have a very different opinion of what constitutes a number or size-controlled regime compared to a moving observer. In theory, scanning dual-polarization weather radar could be used to retrieve DSD dynamics along moving reference frameworks. However, the retrieved DSDs would be affected by large uncertainties, which would make it difficult to derive accurate statistics about the temporal variation in $N_T$ and f(D). Moreover, a second radar would have to be available to rigorously cross-check and confirm all the regimes detected

by the first radar to avoid any false positives.

That being said, the simple detection algorithm proposed in this work is not completely useless. It reliably identifies time intervals with slowly-varying $N_T$ or f(D), as seen by a fixed observer on the ground. Such periods can be used to test the performance and stability of different DSD retrieval algorithms and radar-based rainfall estimation algorithms. The method also gives useful insight into the temporal dynamics of different DSD moments, their co-fluctuation and scaling properties

with respect to the rainfall rate. Such statistics are useful for understanding storm dynamics and improving the modeling of drop size distributions across scales.

In the end, this study serves as a cautionary tale. A testament to how difficult it can be to empirically verify a seemingly easy question. The difficulty of the task also casts doubt on the reliability of previous studies that have claimed to see special regimes based on more rudimentary experimental setups and less rigorous data analyses. Our results show that even with a strict cross-check of disdrometer data, one can easily get fooled into seeing special patterns where there aren't any.

Clearly, more research is needed to understand all the intricacies behind the detection and validation of special rainfall regimes. For example, it would be valuable to also have reliable methods for detecting special regimes from the perspective of a moving observer. The latter is more difficult but also more promising since the detected regimes are more likely to be the result of special microphysical processes. Perhaps a more elaborated technique that combines polarimetric radar with high resolution video-disdrometers could provide the basis for such new detection algorithm at shorter time scales. Future research should also focus on gathering higher-resolution DSD data to shorten the length of the moving time window and better capture the temporal dynamics of DSDs in intense bursts of convective rain. This might unravel some previously unseen number-controlled regimes at higher rainfall intensities. Another interesting line of research could be to consider alternative ways of testing for constant f(D), without having to resort to drop size proxies such as $D_m$. Regardless of the used approach, measurement uncertainty will undoubtedly be a serious issue. Multiple safeguards such as significance testing, co-location and cross-checks between different sensors will have to be put in place to avoid falling into a statistical mirage.

*Data availability.* Both the micro-rain radar data and weather data are part of the Open Data Catalog of KNMI and available for download through the KNMI data platform. The Parsivel disdrometer data is not published yet but are available in monthly NetCDF files upon request to the author.

*Author contributions.* All the work was done by the first author.

*Competing interests.* There are no competing interests.

*Acknowledgements.* This work has been accomplished by using data from the Ruisdael Observatory, a scientific research infrastructure which is partly financed by the Dutch Research Council (NWO, grant number 184.034.015). Special thanks to the Dutch weather service KNMI for providing the Cabauw tower weather data. These data are provided under the OpenData policy of the Dutch government through the KNMI data platform.



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
