# Peer review of "Number and size-controlled rainfall regimes in the Netherlands: physical reality or statistical mirage?"

_EGUsphere, 2023_

## Author Response (AR1)

**Comments Reviewer 1:**

*This is a well written manuscript describing the analysis of Parsivel disdrometer data and micro rain radar (MRR) data to identify and characterize rain regimes known as "number-controlled" or "size controlled". I found the manuscript easy to read and the analysis steps clearly described.*

*I have one question and one recommendation before publishing this manuscript.*

*1. Page 14, Figure 7. What are the vertical lines in Figure 7? Is that another dataset? If it is a dashed line connecting the individual circles, the dashed line is not needed (and is distracting).*

*2. A reviewer pointed out to me many years ago that since analysis of disdrometer data involves discrete measurements with quantized values, the equations should not contain integrals but summations. The integrals are not correct because the drop diameters reach a maximum measured value of Dmax and are not infinite. Thus, I recommend changing the integrals in lines 13 and 14, and in equations (2) and (3) to summations. This would also allow the addition of quantized data into the discussion of error sources.*

**Response to reviewer 1:**

*1. Page 14, Figure 7. What are the vertical lines in Figure 7? Is that another dataset? If it is a dashed line connecting the individual circles, the dashed line is not needed (and is distracting).*

> The lines are used to connect the dots. I have removed them during the revision.

*2. Disdrometer data involve discrete measurements with quantized values. Therefore, the equations should not contain integrals but summations. The integrals are not correct because the drop diameters reach a maximum measured value of Dmax and are not infinite. Thus, I recommend changing the integrals in lines 13 and 14, and in equations (2) and (3) to summations. This would also allow the addition of quantized data into the discussion of error sources.*

> Good point. Indeed, the paper currently does not distinguish between the theoretical quantities expressed as integrals and the sample estimates calculated from discretized disdrometer data. I added some information about this in the revised paper (in the Methodology, in Section 3.4), and provided the equations for explaining how the sample estimates of Nt, Dm and R are derived (using sums rather than integrals).

**Comments Reviewer 2:**

*The Authors proposed a new method to identify special rainfall regimes (i.e. number-controlled and size-controlled regimes). The paper is well written however I have some major issues:*

- *It is not clear the importance/scope of identifying these kind of rainfall regimes. Or in order words, why this study has been performed? Please clarify better in the introduction.*
- *The obtained results show that within the considered dataset the proposed method is not able to identify clearly number-controlled or size-controlled regimes. The latter can be highly due to the fact that the considered precipitation intensities are too low (as stated also by the Author) but can be also highly influenced by the adopted method (section 3.5). The adopted criteria are based on some kind of assumption/considerations? How can the Author*

*be sure that adopting different threshold the results are the same? This can be the key of the problem.*

- *In section 4.6 the Author stated that also the drop fall velocity (v(D)) can have an influence on the results. Which fall velocity has been used to compute the DSD (i.e. measured or theoretical)? Probably the use of experimental/theoretical v(D) relation can help in limit the effect of the fall velocity on the results.*

*MINOR COMMENTS*

1. *Table 1: explain the meaning of Ta, Td, Wa*
2. *Section 4.3: The Z-R relations can be obtained also from disdrometer data in order to see if these results are in agreement with the ones obtained by the method proposed in the paper.*

**Response to reviewer 2:**

*- The importance/scope of identifying these kind of rainfall regimes is not clear. In other words, why was this study performed? Please clarify better in the introduction.*

> Section 1.2 (page 3) already clearly explains why this study has been performed but I agree with the reviewer that the importance/scope of these regimes could still be explained in more detail. I have added a whole paragraph in the Background (Section 1.1) to clarify the relevance of the work.

*- The obtained results show that within the considered dataset the proposed method is not able to identify clearly number-controlled or size-controlled regimes. The latter can be highly due to the fact that the considered precipitation intensities are too low (as stated also by the Author) but can be also highly influenced by the adopted method (section 3.5). The adopted criteria are based on some kind of assumption/considerations? How can the Author be sure that adopting different threshold the results are the same? This can be the key of the problem.*

> All of the assumptions/choices underlying the methodology are critically discussed in Sections 4.4, 4.5 and 4.6. In addition to the presented work, I also clearly state that other thresholds, time windows and statistical metrics were considered. However, in all of these additional experiments, the main conclusion remained the same: there was no compelling evidence of a pure number- or size-controlled rainfall regime in the considered dataset. Obviously, this does not mean that special regimes do not exist. Special meteorological conditions that do not occur in the Netherlands may be required for them to occur. Or perhaps, the dataset is not long enough. On the other hand, the study also shows that it would still be very hard to experimentally confirm such regimes with the help of disdrometers.

*- In section 4.6 the Author stated that also the drop fall velocity v(D) can have an influence on the results. Which fall velocity has been used to compute the DSD (i.e. measured or theoretical)? Probably the use of experimental/theoretical v(D) relation can help in limit the effect of the fall velocity on the results.*

Yes, the fall velocity of the raindrops is a problem. The definition of number and size-controlled regimes implicitly assumes a deterministic relationship between the size and velocity of a raindrop. However, this is hardly the case in reality. This important issue is now explained in full detail in section 3.4, together with the equations used for estimating Nt, Dm and R.

*- Table 1: explain the meaning of Ta, Td, Ws*

They are the air temperature, dew point temperature and wind speed. I have added the information to the caption.

*- Section 4.3: The Z-R relations can be obtained also from disdrometer data in order to see if these results are in agreement with the ones obtained by the method proposed in the paper.*

Good idea, but unfortunately the 1-min sampling resolution of the disdrometer makes it impossible to reliably estimate the Z-R relationship over such short time windows. I can get some estimates but because of the low number of samples, the confidence intervals would be very large.

---

## Author Response (AR2)

**Comments by Referee 2:**

The author addressed all my comments and in my opinion added important information that improve the manuscript. I have just two minor comments to share with the Author. In my opinion the following suggestions can improve the quality of the results but they are not mandatory.

1) Regarding the new section 3.4 added by the Author. I suggest to the author to take a look at the paper of Adirosi et al. (2023) that describes a methodology to improve the quality of disdrometer data starting from the raw data (i.e. Field 93 in the Parsivel2 documentation).

2) Several papers in the literature used 1-minute disdrometer data to obtain Z-R relation. Of course to obtain climatological value of the Z-R coefficient a long time series is needed, however the Z-R relation can be obtained also for each event, although their validity will be more limited.

REFERENCE

Adirosi, E., Porcù, F., Montopoli, M., Baldini, L., Bracci, A., Capozzi, V., Annella, C., Budillon, G., Bucchignani, E., Zollo, A. L., Cazzuli, O., Camisani, G., Bechini, R., Cremonini, R., Antonini, A., Ortolani, A., Melani, S., Valisa, P., and Scapin, S.: Database of the Italian disdrometer network, Earth Syst. Sci. Data, 15, 2417–2429, https://doi.org/10.5194/essd-15-2417-2023, 2023.

**Point by point response:**

1) Regarding the new section 3.4 added by the Author. I suggest to the author to take a look at the paper of Adirosi et al. (2023) that describes a methodology to improve the quality of disdrometer data starting from the raw data (i.e. Field 93 in the Parsivel2 documentation).

**Response:** Thanks for sharing the reference but I do not see any valid reason to cite this paper within the context of my work. I categorically refuse requests to cite papers that do not meaningfully contribute to the topic, and do not help improve the scientific quality of the work. In my opinion, the question of how to process raw DSD data from Parsivel optical disdrometers is not the main focus of this paper. Many other, more influential papers about this topic are already available but I see no valid reason to cite either of them.

2) Several papers in the literature used 1-minute disdrometer data to obtain Z-R relation. Of course to obtain climatological value of the Z-R coefficient a long time series is needed, however the Z-R relation can be obtained also for each event, although their validity will be more limited.

Response:

Thanks for the comment. I fully agree. The paper already contains a sentence that conveys this idea (see line 60):

*"The application of this technique can, however, be rather challenging. The substantial measurement uncertainty that impacts Z-R measurements, along with the reliance on strong modeling assumptions (specifically, the fact that Z-R relationships can be approximated by a power-law) imply that large sample sizes are needed to reliably estimate $\alpha$ and $\beta$."*